# Tunable and parabolic piezoelectricity in hafnia under epitaxial strain

Hao Cheng [1,2,4], Peijie Jiao[1,2,4], Jian Wang[1,2], Mingkai Qing[1,2], Yu Deng[1,2], Jun-Ming Liu [1], Laurent Bellaiche [3] ✉, Di Wu [1,2] ✉ & Yurong Yang [1,2] ✉

Piezoelectrics are a class of functional materials that have been extensively used for application in modern electro-mechanical and mechatronics technologies. The sign of longitudinal piezoelectric coefficients is typically positive but recently a few ferroelectrics, such as ferroelectric polymer poly(vinylidene fluoride) and van der Waals ferroelectric $CuInP_2S_6$, were experimentally found to have negative piezoelectricity. Here, using first-principles calculation and measurements, we show that the sign of the longitudinal linear piezoelectric coefficient of $HfO_2$ can be tuned from positive to negative via epitaxial strain. Nonlinear and even parabolic piezoelectric behaviors are further found at tensile epitaxial strain. This parabolic piezoelectric behavior implies that the polarization decreases when increasing the magnitude of either compressive or tensile longitudinal strain, or, equivalently, that the strain increases when increasing the magnitude of electric field being either parallel or antiparallel to the direction of polarization. The unusual piezoelectric effects are from the chemical coordination of the active oxygen atoms. These striking piezoelectric features of positive and negative sign, as well as linear and parabolical behaviors, expand the current knowledge in piezoelectricity and broaden the potential of piezoelectric applications towards electro-mechanical and communications technology.

Piezoelectricity describes the conversion from electrical energy to mechanical energy and vice versa. It has been a subject of extensive research as piezoelectrics serve as critical components in many modern devices ranging from sonar, medical ultrasound, sensors, actuators, and vibration-powered electronics[1–5]. Piezoelectricity is quantified by piezoelectric coefficients, which characterize how the polarization changes in response to a strain $\varepsilon$ ($\frac{\partial P}{\partial \varepsilon}$, piezoelectric stress coefficient $e$) or stress $\sigma$ ($\frac{\partial P}{\partial \sigma}$, piezoelectric stress coefficient $d$). Most piezoelectric materials, such as ferroelectric perovskites, possess *positive* longitudinal piezoelectric coefficient where the lattice expands along the direction of the applied external electric field[6–10] (see Fig. 1a). Recently, a *negative* longitudinal piezoelectric coefficient

has been experimentally observed in low-dimensional ferroelectric polymer poly(vinylidene fluoride) (PVDF)[11] and van der Waals ferroelectric $CuInP_2S_6$[12,13], where the lattice contracts along the direction of an applied electric field (see Fig. 1b). The origin of the counterintuitive negative longitudinal piezoelectric effect in PVDF resides in the unique microstructures with intermixed crystalline lamellae and amorphous regions[11]. In $CuInP_2S_6$, negative longitudinal piezoelectricity is attributed to the coupling between the large displacement of Cu ions and the reduced lattice dimensionality under the electric field[12]. The negative longitudinal piezoelectricity of both PVDF and $CuInP_2S_6$ is related to the reduced lattice dimensionality. Negative longitudinal piezoelectricity has also been predicted by first principles in zinc

[1]Laboratory of Solid State Microstructures, Nanjing University, Nanjing 210093, China. [2]Jiangsu Key Laboratory of Artificial Functional Materials, Department of Materials Science and Engineering, Nanjing University, Nanjing 210093, China. [3]Physics Department, Institute for Nanoscience and Engineering, University of Arkansas, Fayetteville, AR 72701, USA. [4]These authors contributed equally: Hao Cheng, Peijie Jiao. ✉e-mail: laurent@uark.edu; diwu@nju.edu.cn; yangyr@nju.edu.cn

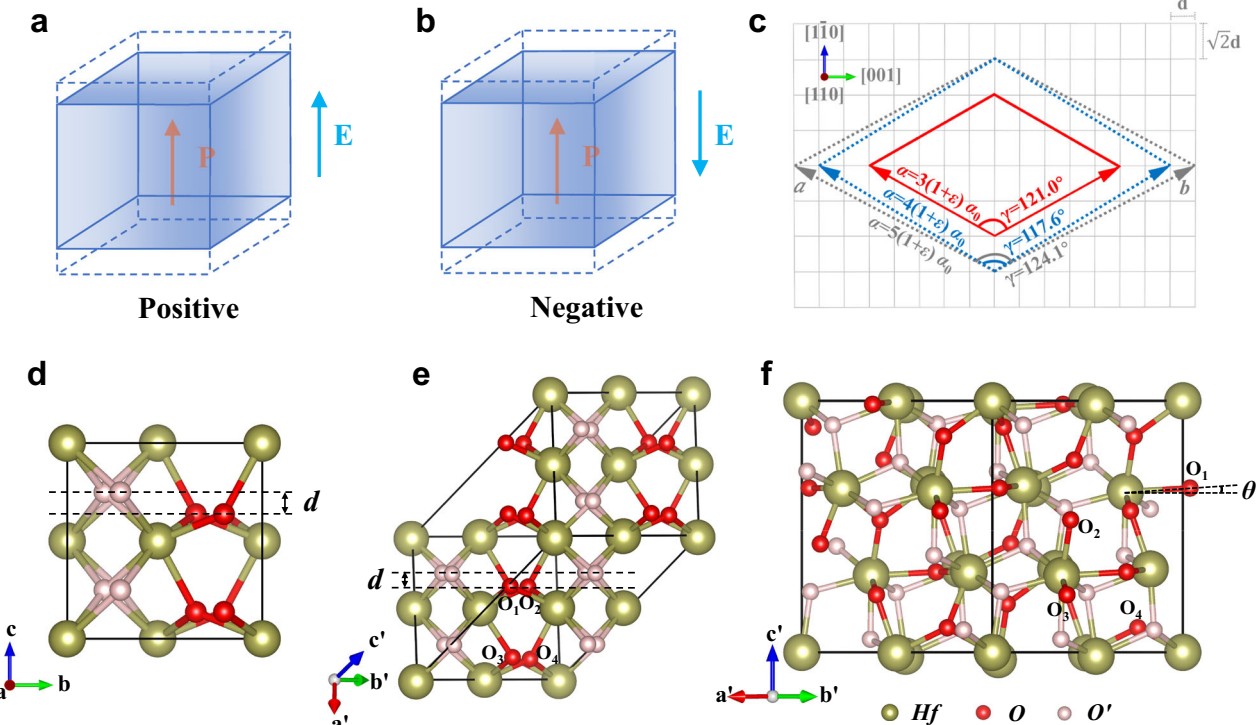

**Fig. 1 | Piezoelectric effect and structures of ferroelectric $Pca2_1$ and $Pca2_1$-like HfO$_2$.** Schematic diagram for the **a** positive and **b** negative piezoelectric effect. **c** Sketch of lattices matching between (111)-oriented hafnia and (110)-oriented substrate. The gray grids represent the substrate. Red solid, blue dashed, and gray dashed vectors represent the possible lattice vectors of (111)-oriented hafnia. The in-plane lattice angle γ and lattice amplitude $a$ are given. **d** Structure of the orthorhombic $Pca2_1$ phase. (111)-oriented $Pca2_1$-like structures viewed (**e**) in the same direction as that in **d** and **f** in a direction that is perpendicular to the $c'$ axis along the [111] direction. The larger spheres represent the Hf atoms. The red and pink smaller spheres represent O atoms that contribute to polarization and do not contribute to polarization, respectively. The oxygen atoms shown via red spheres and contributing to the polarization can be divided into four categories: O$_1$, O$_2$, O$_3$, and O$_4$. θ in **f** represents the angle between O$_1$-Hf bond and the $a'$–$b'$ plane.

blende[14], wurtzite[15], hexagonal ABC ferroelectrics[16], and van der Waals layered solids BiTeX[17]. Though both negative and positive signs of linear piezoelectricity have been predicted and observed in different structures, a quadratic term can also dominate the piezoelectric response in certain materials[18].

Very interestingly, it has been reported recently that hafnia (HfO$_2$) exhibits negative longitudinal piezoelectricity[19–22] while positive longitudinal piezoelectricity was also observed in HfO$_2$ thin films[23–27]. It is thus an open question of why hafnia could exhibit both positive and negative longitudinal piezoelectricity. Though first-principles calculations provide an atomic mechanism to explain negative piezoelectricity there[21], the origin of the positive piezoelectricity is unknown. It is also legitimate to wonder if the magnitude and sign of linear and quadratic piezoelectric effects for hafnia can be tuned, for example, by epitaxial strain—which has been used to tune ferroelectricity in various compounds[28–31].

Here we reveal from first-principles calculations and measurements that both the sign and magnitude of the longitudinal piezoelectricity of HfO$_2$-based films are tunable by strain. Different from the previous work[21], our calculations are conducted on (111)-oriented thin films, which are favorable in experiments with compatibility existing between the substrates and the thin films. We also show that the piezoelectric behavior is dominated by both linear and quadratic piezoelectric coefficients. At negative and small tensile epitaxial strains, the piezoelectric coefficient mostly arises from the linear coefficient, which is negative. At large tensile epitaxial strain, the coexistence of a positive linear coefficient and a negative quadratic term leads to an electrical polarization being nonlinearly dependent on the longitudinal strain. Furthermore, at the critical intermediate epitaxial strain for which the linear coefficient is basically annihilated, the

piezoelectric behavior is parabolic, where the polarization decreases when increasing the magnitude of either the tensile or compressive longitudinal strains, or the lattice expands when increasing the magnitude of applied external electric field being either parallel or anti-parallel to the direction of polarization. Significantly, we elucidate the piezoelectric phenomenon by examining atomic displacements at the microscopic level. Our calculations agree with the experimental results.

## Results

### Structure under (111) epitaxial strain

Let us first investigate structures and properties predicted by simulations. (111)-oriented ferroelectric HfO$_2$-based films can be grown on (001) and (110) oxides substrate. Here, we construct the (111) film structures under epitaxial strain, where the in-plane lattice constants are fixed and equal to each other $a = b$, the angle γ between the in-plane lattice constants is fixed to be close to 120°. Figure 1c shows the possible (111)-oriented HfO$_2$ matching for the (110)-oriented substrate. The value of $a$ (or $b$) determines the amount of epitaxial strain, and the value of γ determines the shear strain (see Fig. 1c and Table 1). Comparing the epitaxial strain and shear strain shown in Table 1, the lattice vector of (111)-HfO$_2$ films with magnitude of $a = 3a_0$ and γ = 121° is the most possible to match the (110) substrate. We consider the (111)-HfO$_2$ films in monoclinic $P2_1/c$ phase, orthorhombic $Pca2_1$ phase, tetragonal $P4_2/nmc$ phase, and another orthorhombic $Pmn2_1$ phase, and rhombohedral phases, as these phases were experimentally synthesized in thin films[32–49]. The symmetry of these states will change when going from phases in bulk to (111)-oriented structures with fixing in-plane lattice constants $a = b$ and γ ≈ 120°. That is why we use the notations $Pca2_1$-like, $P4_2/nmc$-like, and $Pmn2_1$-like to represent the (111)-oriented

structure of the corresponding phases. The rhombohedral of $R3m$ and $R3$ phases[38,49] are considered in our calculations. However, the $R3m$ phase relaxes to a $P42/nmc$-like phase, and the $R3$ phase turns to a $Pca2_1$-like phase under the strain considered here. Figure 1d–f show the atomic structures of the original $Pca2_1$ phase without any constraints and the constrained (111)-oriented $Pca2_1$-like structure with lattice vector $c'$ along the [111] direction and lattice constants $a = b$.

We first determine the structural phases under epitaxial strains. Figure 2a displays the energies of the three studied phases under [111] epitaxial strain. The lowest energy of the (111)-oriented structures is that of $Pca2_1$-like phase with in-plane lattice constants $a_O = b_O = 7.236$ Å where the epitaxial strain is zero. The $Pca2_1$-like phase for the epitaxial strains range from −3% to 4.5% has lower energy than $P4_2/nmc$-like-t, and $Pmn2_1$-like phases. For compressive strains with a magnitude larger than 3%, the ground phase becomes a monoclinic phase $P4_2/nmc$-like-m where the lattice angles significantly change as compared to tetragonal $P4_2/nmc$-like-t. On the other hand, for tensile strains larger than 1.5%, the energy of the $P2_1/c$-like phase has lower energy than the $Pca2_1$-like phase and becomes the ground state (see Fig. S1). The energy difference between the $P2_1/c$-like phase and the $Pca2_1$-like phase under tensile strain is comparable to that of these phases in bulk[50]. In our experiments, $Pca2_1$-like rather than $P2_1/c$-like phase is grown successfully, though the $Pca2_1$-like phase is higher energy than $P2_1/c$-like phase. This may come from surface and/or interface effects, which make the $Pca2_1$-like phase more stable. We therefore focus on the piezoelectric response of $Pca2_1$-like phase from −3% to 4.5%.

### Piezoelectric behavior of (111) HfO₂-based thin films

Figure 2b shows the out-of-plane polarization as a function of out-of-plane (longitudinal) strain at the zero epitaxial strain. Different from the piezoelectric response of usual piezoelectric perovskites in which the polarization is linear with strain, the polarization in (111)-oriented HfO₂ exhibits an obviously nonlinear phenomenon with out-of-plane strain. This implies that linear and quadratic piezoelectric terms should both be included

$$\Delta P_3 = e_{33}\varepsilon_3 + \frac{1}{2}B_{333}\varepsilon_3^2, \tag{1}$$

where the number 3 indicates the out-of-plane direction, $P$ is the polarization, $\varepsilon$ is the strain, $e$ is the linear piezoelectric coefficient, and $B$ is the quadratic piezoelectric coefficient. As shown in Fig. 2b, Eq. (1) fits very well the polarization-versus-strain curve, with $e_{33} = -0.15$ C/m²

**Table 1 | Epitaxial strains of (111)-oriented HfO₂ films on (110)-oriented substrates of LaAlO₃ (LAO), (LaAlO₃)₀.₃₃-(Sr₂AlTaO₆)₀.₆₇ (LSAT), and SrTiO₃ (STO)**

|  | LAO | LSAT | STO |
|---|---|---|---|
| $a_{sub}$(Å) | 3.771 | 3.868 | 3.908 |
| $3a_O$ (121.0°) | −0.2% | 2.4% | 3.4% |
| $4a_O$ (117.6°) | 6.6% | 9.4% | 10.5% |
| $5a_O$ (124.1°) | −5.6% | −3.2% | −2.2% |

In the first column, $a_O$ represents the lattice constant (7.236 Å) of hafnia film at zero strain and the angle in parentheses represents the in-plane lattice angle.

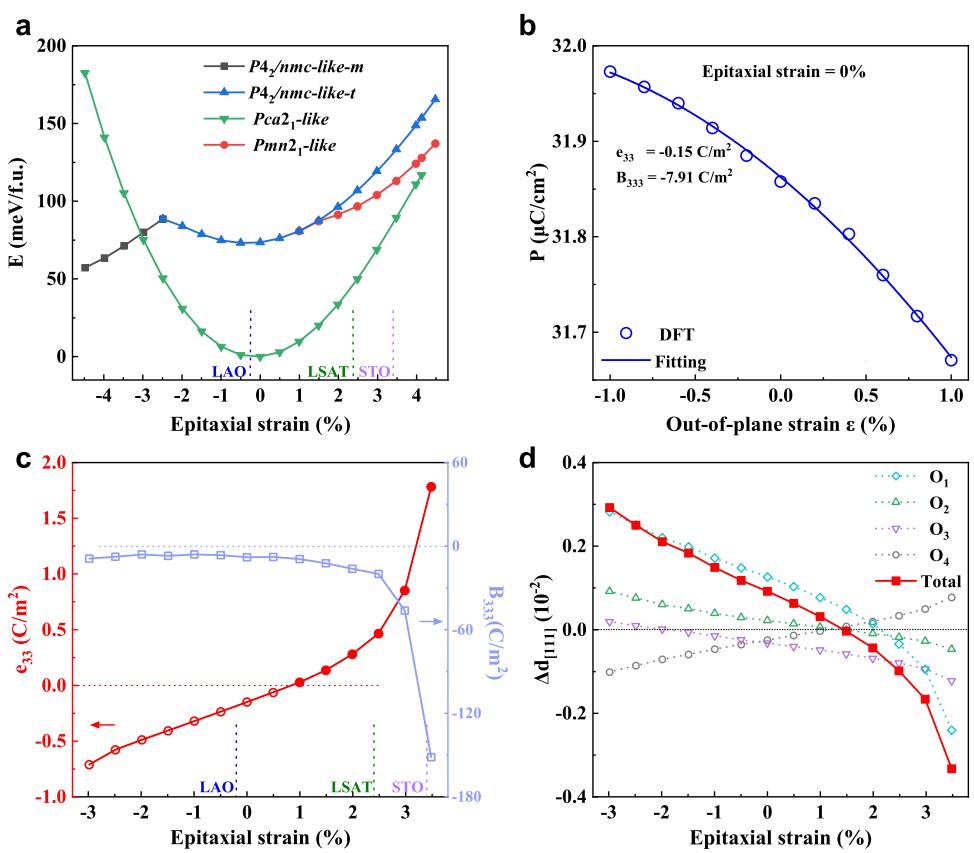

**Fig. 2 | Phase stability and piezoelectric response of HfO₂ under [111] epitaxial strain. a** Energies of different phases as a function of the epitaxial strain. **b** The polarization as a function of out-of-plane strain at the epitaxial strain of zero. **c** The linear piezoelectric coefficients $e_{33}$ and quadratic coefficient $B_{333}$. **d** The displacements (Δd) of the four types of oxygen ions contributing to the polarization along the [111] direction under an out-of-plane tensile strain of 1% (which corresponds to the difference of fractional coordinates as compared to the zero strain, $\Delta d_{[111]} = d_{[111],1\%} - d_{[111],0\%}$) at different epitaxial strained films. The blue, green, and purple vertical dashed lines in panels a and c represent the epitaxial strains corresponding to LAO, LSAT, and STO substrates, respectively.

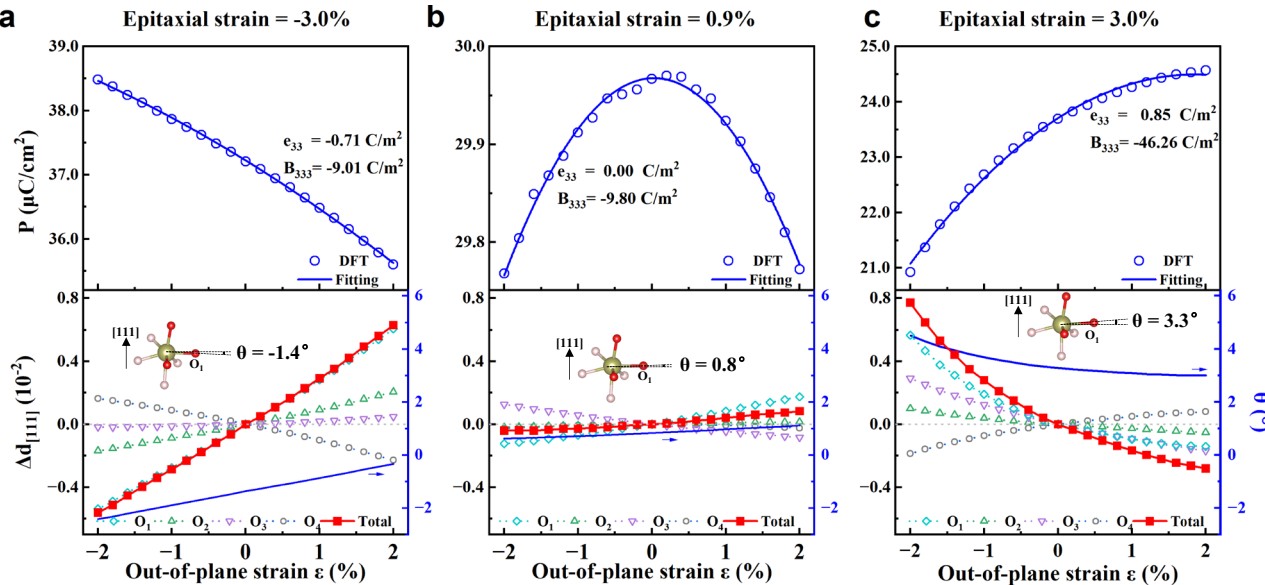

**Fig. 3 | Piezoelectric responses under longitudinal strain.** The polarization and the displacement $\Delta d$ along [111] direction (the difference of fractional coordinates between the strained and zero strain films, $\Delta d_{[111]} = d_{[111],\varepsilon} - d_{[111],0\%}$) of the four kinds of O ions contributing to polarization as a function of out-of-plane strain for epitaxial strains of **a** –3.0%, **b** 0.9%, and **c** 3%. The insets in the bottom panels show the local configurations around $O_1$. The angle of $O_1$-Hf bond and $a'$-$b'$ plane is –1.4°, 0.8°, and 3.3° at zero out-of-plane strain in **a**–**c**, respectively.

and $B_{333} = -7.91$ C/m². Note the converse piezoelectric effect including linear and quadratic coefficients is also provided in supplemental information as equation S2.

Figure 2c shows $e_{33}$ and $B_{333}$ as a function of epitaxial strain. For epitaxial strains ranging from −3% to 2.5%, $e_{33}$ approximately linearly increases from negative −0.71 C/m² to positive 0.47 C/m². On the other hand, $B_{333}$ is always negative within this epitaxial strain range. The epitaxial strain thus induced an inversion of sign from negative to positive for $e_{33}$, which along a nonnegligible quadratic piezoelectric coefficient, can lead to unusual piezoelectric phenomena. For instance, at compressive epitaxial strain (−3%, for example), both $e_{33}$ and $B_{333}$ are *negative*, and the polarization basically linearly decreases with the increase of out-of-plane strain (see top panel of Fig. 3a). At small tensile epitaxial strain (0.9%, for example), $e_{33}$ is very small, and $B_{333}$ thus dominates the polarization response to the longitudinal strain. As shown in the top panel of Fig. 3b, the polarization thus displays a parabolic behavior with respect to the out-of-plane strain $\varepsilon$. As a result, when the out-of-plane strain $\varepsilon<0$ (see top panel of Fig. 3b), the polarization decreases with the increase of magnitude of compressive strain, showing a *positive* piezoelectricity. In contrast, when $\varepsilon>0$ (see top panel of Fig. 3b), the polarization decreases with the increase of magnitude of tensile strain, showing a *negative* piezoelectricity. Furthermore, when the epitaxial strain is larger than 0.9%, $e_{33}$ becomes *positive*. Moreover, for epitaxial strains larger than 3.0%, the linear $e_{33}$ and quadratic $B_{333}$ are both large while possessing opposite sign. Consequently, they lead to a strongly nonlinear piezoelectric behavior (as shown in the top panel of Fig. 3c), polarization obviously nonlinearly increases when the out-of-plane strain increases, exhibiting *positive* piezoelectricity, different from that at compressive epitaxial strain that shows *negative* piezoelectricity (see top panel of Fig. 3a), and also different from that at epitaxial strains close to 0.9% which can exhibit either *positive* or *negative* piezoelectricity depending on the sign of the out-of-plane strain (see top panel of Fig. 3b). Notably, the parabolic piezoelectricity is unveiled in hafnia at small tensile epitaxial strain, thereby advancing the current understanding of the field of piezoelectricity.

To determine the microscopic origin of these striking different piezoelectric phenomena for different epitaxial strains, we investigate the atomic structural response to out-of-plane strain. Figure 1e, f shows

the atomic structure of the $Pca2_1$-like phase with the $c'$ lattice vector being along the [111] direction. Similar to the original unit cell of the orthorhombic $Pca2_1$ phase (see Fig. 1d) where the polarization comes from the displacement of certain oxygen ions, the polarization of the (111)-oriented $Pca2_1$-like phase mainly originates from the displacements (the projection of $d$ along [111] direction, $d_{[111]}$) of the four kinds of oxygen ions ($O_1$, $O_2$, $O_3$, and $O_4$ shown in Fig. 1e, f) with respect to the high symmetry tetragonal and paraelectric $P4_2/nmc$ phase. The displacements along the [111] direction ($\Delta d_{[111]}$) for these four kinds of oxygen ions under an out-of-plane strain of 1% (as compared to displacements for zero out-of-plane strain, $\Delta d_{[111]} = d_{[111],1\%} - d_{[111],0\%}$) are related to the change of polarization and the piezoelectric response. Figure 2d shows these displacements ($\Delta d_{[111]}$) for different epitaxial strains. All the four kinds of oxygen ions significantly move under out-of-plane tensile strains. $O_1$, $O_2$, and $O_3$ have positive displacements at the epitaxial strain of −3%, which decrease when the epitaxial strain increases from −3% to 3.5%, leading to the inversion of these displacements from positive to negative. The displacement of $O_1$ under large compressive or tensile epitaxial strains dominates the total displacement. Different from the other three kinds of oxygen ions, $O_4$ has a negative displacement at the epitaxial strain of −3%, which increases with the enhancement of the epitaxial strain from −3% to 3.5%, leading to the inversion of displacements from negative to positive. As the total displacements of $O_2$ and $O_3$ nearly cancel that from $O_4$, the total displacement of four kinds of oxygen ions behaves similarly to that of $O_1$ (see Fig. 2d). The total displacement thus decreases when the epitaxial strain changes from −3% to 3.5%. The displacement of the four types of O atoms is very small around the epitaxial strain of 1.5%, which means that their contribution to polarization change is very small, resulting in minimal changes in overall polarization. It is *positive* for epitaxial strains smaller than 1.5%, *negative* for epitaxial strains larger than 1.5%, and almost 0 at the strain of 1.5%. This critical strain of 1.5% is relatively close to that of 0.9% for which $e_{33}$ changes from *negative* to *positive* (the small difference between the two critical strain results from the fact that the critical strain of 1.5% is extracted by using Born effective charges for which only the ionic contribution to polarization is included, while 0.9% is obtained by employing Berry phases methods considering both ionic and electronic contributions to polarization). As shown in Fig. 2c and d, the *positive* (*negative*) total

displacements of oxygen ions are thus rather consistent with the *negative* (*positive*) $e_{33}$.

We then consider the piezoelectric response in more detail. Figure 3 shows the polarization and $\Delta d_{[111]}$ of oxygen ions as functions of applied out-of-plane strain $\varepsilon$ for epitaxial strains of −3.0%, 0.9%, and 3.0%. $\Delta d_{[111]}$ for the four kinds of oxygen ions ($O_1$, $O_2$, $O_3$, and $O_4$) exhibit different signs and magnitude with respect to each other, as a function of the out-of-plane strain. At the epitaxial strain of −3%, the displacements of $O_1$, $O_2$, and $O_3$, which are negative at negative out-of-plane strains, increase with strain and become positive when the out-of-plane strain is positive. On the other hand, the displacements of $O_4$, which are positive at negative out-of-plane strains, decrease with strain and become negative when the strain is positive. The total displacements of the four kinds of oxygen ions are the main contribution to the polarization (though the electron contribution is also large for some cases) and lead to the corresponding behavior of polarization with respect to the out-of-plane strain. At the epitaxial strain of −3%, the almost linear polarization dependence on the out-of-plane strain results from the linear displacements of the oxygen ions. At the large tensile epitaxial strain of 3.0%, the polarization is nonlinearly dependent on the out-of-plane strain, in line with the concomitant nonlinear total displacement of oxygen ions. At the epitaxial strain of −0.9%, the polarization has the shape of a parabola while oxygen ions displacements appear to be all linear with respect to out-of-plane strain. This is because electrons largely contribute to the polarization in that case. Note that the aforementioned linear, nonlinear, and parabolical piezoelectric behaviors can be explained by the linear and quadratic piezoelectric terms of Eq. (1). Overall, the unusual piezoelectric effect in hafnia is due to the competitive relationship of the four kinds of oxygen atoms on the displacement under epitaxial strain.

Comparing to the displacements ($\Delta d_{[111]}$) coming from $O_1$, $O_2$, $O_3$, and $O_4$ under out-of-plane strain, as shown in the bottom panels of Fig. 3, $\Delta d_{[111]}$ from $O_1$ is the main contributor to the total displacement and the piezoelectric response. The local atomic configuration around $O_1$ shows in the insets in the bottom panels of Fig. 3, where $O_1$-Hf bonds are close to being perpendicular to the out-of-plane ([111]) direction. This configuration leads to $O_1$ being easier to move under out-of-plane strain than other oxygen ions. At the epitaxial strain of −3%, $\theta = -1.4°$ ($\theta$ is the angle between the $O_1$-Hf bonds and the $a' - b'$ plane), and the magnitude of $\theta$ becomes larger under compressive strain ($O_1$ moves antiparallel to polarization and polarization increases), resulting in a negative piezoelectricity. Conversely, at the epitaxial strain of 3%, $\theta = 3.3°$ and $\theta$ becomes larger under compressive strain ($O_1$ moves parallel to polarization and polarization decreases), inducing a positive piezoelectricity. At the epitaxial strain 0.9%, $\theta$ and the coordinate of $O_1$ do not change much under out-of-plane strain, the electronic contribution dominates to change of polarization, which gives rise to a parabolic piezoelectricity.

## Experimental confirmation

The epitaxial strain-induced inversion of piezoelectricity from positive to negative sign is confirmed by our experiments. (110)-oriented LaAlO$_3$ (LAO), (LaAlO$_3$)$_{0.33}$-(Sr$_2$AlTaO$_6$)$_{0.67}$ (LSAT), and SrTiO$_3$ (STO) substrates are considered. To reveal the lattices matching between the (111)-oriented HfO$_2$ film and the (110)-oriented substrate, Fig. 1c shows the possible matching with relatively small lattice constants $a$ and in-plane lattice angle $\gamma$ for HfO$_2$. Table 1 displays the possible epitaxial strains for (111)-oriented HfO$_2$ on (110)-oriented LAO, LSAT, and STO substrates. Considering the magnitude of epitaxial strain (including in-plane shear strain), the lattice vector of HfO$_2$ of about $3a_0$ ($a_0 = 7.236$ Å, the lattice constant of (111)-oriented HfO$_2$ at zero strain) and in-plane angle $\gamma = 121°$ form the most probable matching in experiments (Fig. 1c). Figure 4a, b display the aomic resolution of High-angle annular dark-field scanning transmission electron microscopy (HAADF-STEM) images of HfO$_2$-based films (Hf$_{0.5}$Zr$_{0.5}$O$_2$, HZO) on STO

substrate with this lattice match scenario, and Table 1 shows that the possible epitaxial strain of Hf$_{0.5}$Zr$_{0.5}$O$_2$ on STO, LSAT, and LAO is 3.4%, 2.4%, and −0.2%, respectively.

Figure 4c shows X-ray diffraction (XRD) θ-2θ patterns of HZO/LSMO heterostructures deposited on (110)-oriented LAO, LSAT, and STO substrates. The thicknesses of LSMO and HZO layers were kept at 20 nm and 10 nm, respectively. With increasing the lattice constant of the substrate, the (110) peak from LSMO shifts to higher angles, indicating the compression of the LSMO lattice along the out-of-plane direction. The main feature of HZO films for all these samples is a peak at around 30.2° from the (111) plane of o-HZO. To experimentally determine the sign of the piezoelectric coefficient in HZO films, we carry out measurements by means of piezoresponse force microscopy (PFM). In a material with a positive longitudinal piezoelectric coefficient, the sample oscillation and the driving electric field will be in phase detected by PFM when the polarization is oriented downward, while it will be in anti-phase when the polarization is oriented upward[21,51]. The PFM phase loop is directly related to the sign of the piezoelectricity, and the scenario has been confirmed for PVDF, which is a well-known negative piezoelectric material[11]. In Fig. 4d, a clockwise rotation of the PFM phase signal is indicative of a *positive* piezoelectricity in the HZO film deposited on STO and LSAT substrates. In this case, the oscillation is in phase (anti-phase) with the ac modulation applied on the tip at the large enough positive (negative) dc bias, which aligns the polarization downward (upward). In contrast, as shown in Fig. 4d, in the HZO film deposited on LAO substrate, the phase loop is anti-clockwise, indicative of a *negative* piezoelectricity. The measurements of positive and negative piezoelectricity on substrates LAO, LSAT, and STO are consistent very well with our calculations (Fig. 2c).

Consequently, the buffer LSMO layer with different thicknesses, which is distinctly relaxed on the LAO substrate, provides different epitaxial strains for HZO thin films. A series of HZO/LSMO heterostructures with identical thicknesses (10 nm) of HZO and different thicknesses of LSMO (20-66 nm) were synthesized on LAO substrates. XRD θ-2θ scans of these samples are displayed in Fig. 5a. The peak shift of LSMO with increasing thickness from 20 to 66 nm indicates the gradual strain relaxation of the interface LSMO layer. Figure 5c−e shows reciprocal space mappings around the (310) spot of LAO for HZO thin films deposited on 20-, 33-, and 66-nm-thick LSMO buffer layers, respectively. It is observed that the (310) spot of LSMO shifts toward the upper left with increasing thickness, indicating the relaxation with increasing in-plane (and concomitantly decreasing out-of-plane) lattice constants. This allows us to continuously tune the epitaxial strain on HZO from a compressive to a tensile value, which arises from fully relaxed buffer layers. Figure 5b shows the PFM phase loops of the HZO films deposited on 20-, 33- and 66-nm-thick LSMO buffered LAO substrates. For the thinner buffer of 20 nm and 33 nm, the PFM shows a negative sign for the piezoelectric coefficient, while for the 66 nm buffer of LSMO (which is fully relaxed and provides a tensile strain for HZO), the piezoelectric coefficient adopts a positive sign.

In order to achieve a dynamic transition between positive and negative piezoelectric coefficients, one way is to transfer the HfO$_2$-base thin film to the organic substrate polyethylenenaphthalate (PEN) and use epoxy as the glue to transfer strain to the HfO$_2$-base thin film[52]. Without strain, the piezoelectric coefficient of the HfO$_2$ thin film on PEN is negative (see Fig. 2 and Fig. S5), while it is positive when the HfO$_2$ thin film is under tensile strain by stretching epoxy.

## Frequency conversion induced by the parabolic piezoelectric effect

We then study the piezoelectric response under AC sinusoidal electric field with a frequency of 10 GHz by molecular dynamics simulation with machine learning atomic interaction potential[53–55]. Figure 6 shows the out-of-plane lattice constant changes of (111)-oriented HfO$_2$ film for

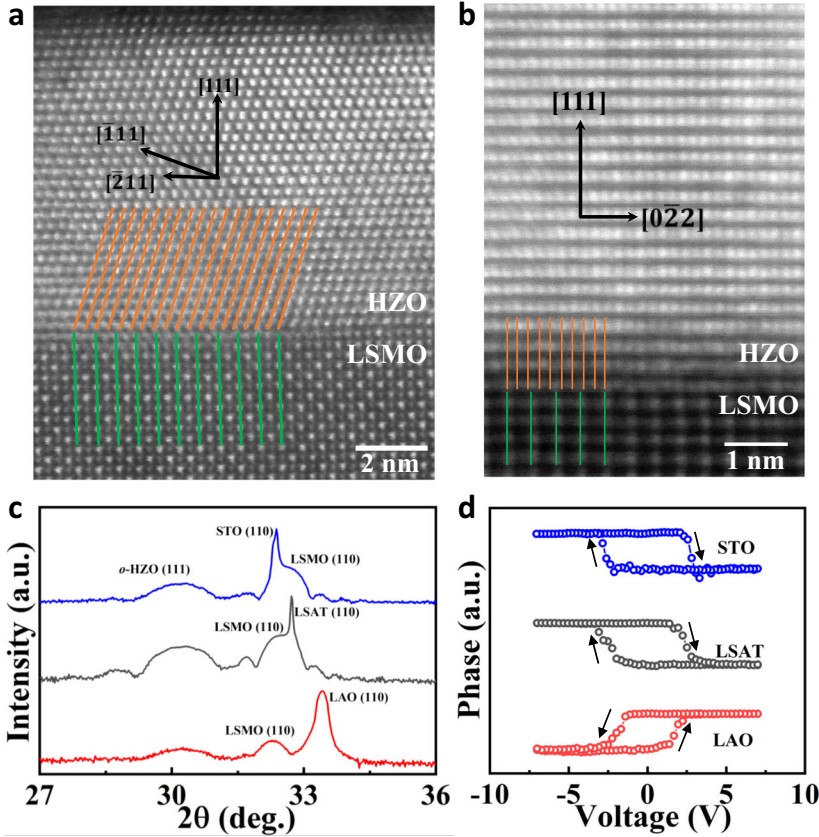

**Fig. 4 | Structure and piezoelectricity of [111]-oriented HZO films grown on (110) perovskite substrates.** Atomic resolution HAADF-STEM images of the HZO/LSMO film observed along **a** [001] and **b** [1–10] zone axis of STO. **c** XRD θ-2θ patterns of HZO/LSMO heterostructures deposited on (110)-oriented LAO, LSAT, and STO substrates. **d** PFM phase loops of the HZO films deposited on LAO, LSAT, and STO substrates.

epitaxial strains of 2.5%, −3.5%, and 0.9% under an AC field oriented along the [111] direction. For the tensile epitaxial strain of 2.5% for which $HfO_2$ film adopts positive piezoelectricity, the change of lattice constant exhibits a *sine* function that is similar to that of the AC field as function of time (Fig. 6b). In contrast, for a compressive epitaxial strain of −3.5% for which the film possesses negative piezoelectricity, one can see a *cosine* shape for the lattice constant's change (Fig. 6c). Piezoelectric response of the film for both the epitaxial strain of 2.5% and −3.5% possesses the same frequency with the AC field. On the other hand, the film at the epitaxial strain of 0.9% displays a very different response, which shows only positive strain under field and has a frequency of two times that of the AC field (Fig. 6d). This behavior is induced by the parabolic piezoelectric response where the lattice expands when increasing the magnitude of electric field being either parallel or antiparallel to the polarization (Fig. 3b or Fig. S2). This conversion effect of AC field frequency by piezoelectricity has never been discovered before, to the best of our knowledge, which can be used as an amplifier of frequency in communications technology.

## Discussion

Linear positive and negative piezoelectricity are observed in $HfO_2$-based films and can be tuned by epitaxial strain. Nonlinear and even parabolical piezoelectric behavior are revealed which originates from a nonnegligible quadratic piezoelectric coefficient. These unusual piezoelectric phenomena expand the concept of piezoelectricity and broaden their potential for electro-mechanical and mechatronics devices. We thus hope the presently discovered piezoelectricity, especially that associated with the amplifier of frequency, will be discovered in more materials and will be taken advantage of in applications.

For nanometric thicknesses and nontoxic nature, $HfO_2$-based piezoelectric film can be used in medical devices for diagnoses inside the human body, and are good candidates for implantable cardio-mechanical electric sensor applications with limited piezoelectricity. Also, as $HfO_2$ thin films can be grown on silicon and are compatible with the electronics industry, they are also good candidates for piezoelectric microelectromechanical systems (MEMS) in nanoscale.

## Methods

### First-principles simulations

The DFT calculations are performed using the Vienna Ab initio Simulation Package (VASP) code[56,57]. We use the PBEsol function[58] based on projector-augmented wave method[59]. The energy cutoff was 550 eV and the Hellman-Feynman forces are taken to be converged when they become smaller than 1 meV/Å on each atom for atomic structure relaxation. For the (111)-oriented hafnia, a supercell containing 36 atoms was used for the calculations.

Molecular dynamics (MD) simulations with the machine learning force field (MLFF) were performed to study the piezoelectric response under AC field[53–55]. A supercell containing 144 atoms was employed within the NPT ensemble using a Langevin thermostat[60], while the MD time step was set to 1 fs. The temperature was confined to 10 K to minimize the thermal fluctuation noise. The root mean square errors of the MLFFs from the training data sets were less than 0.29 meV/atom in energy, 0.07 eV/Å in force, and 0.8 kbar in stress tensors. These root mean square errors are similar to those in previous MLFF simulation[54,61,62]. The scheme of the electric enthalpy functional is employed to determine the response to finite electric fields[63,64]. The polarization was calculated by the Berry Phase method[65] and the product of the atomic displacements with the Born effective charges[66].

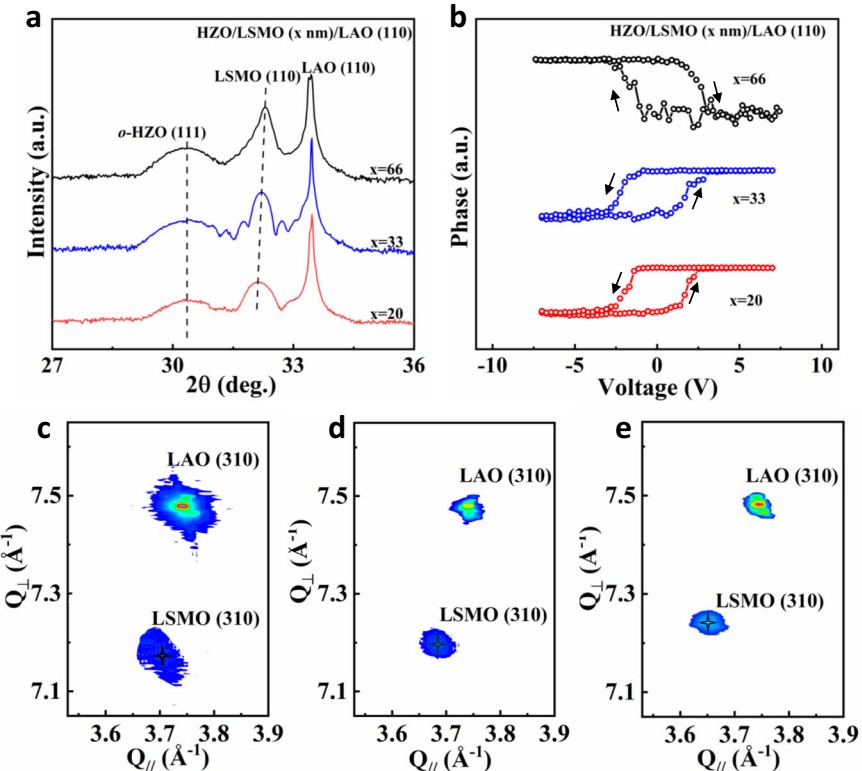

**Fig. 5 | [111]-oriented HZO films grown on (110) LAO substrates with different thicknesses of LSMO buffer layers. a** XRD θ-2θ patterns of HZO films deposited on (110)-oriented LAO with 20-, 33-, 66-nm-thick LSMO buffer layers. **b** PFM phase loops of the HZO films deposited on 20-, 33- and 66-nm-thick LSMO buffered LAO substrates. **c–e** Reciprocal space mappings around the (310) spot of HZO/LSMO/LAO with **c** 20-, **d** 33-, and **e** 66-nm-thick LSMO buffer layers.

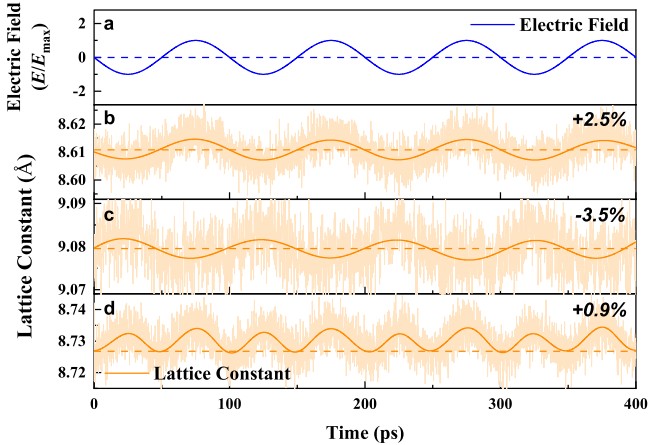

**Fig. 6 | Piezoelectric response under sinusoidal electric field applied along the out-of-plane direction from MD simulations. a** The sinusoidal electric field $E(t) = E_{max}\sin(2\pi\omega t + \pi)$, where $t$ is time, $\omega$ is the frequency which is 10 GHz here, $E_{max}$ is the maximum amplitude of the electric field, which is 4 MV/cm (for **b**, **c**), or 8 MV/cm (for **d**) to make the piezoelectric effect more clearly. **b–d** The out-of-plane lattice constant of the [111]-oriented $HfO_2$ for epitaxial strain of (**b**) 2.5%, (**c**) −3.5%, and (**d**) 0.9%. The high-frequency noise of light-orange lines in the MD simulations is filtered by fast Fourier transform (FFT) smoothing as dark-orange solid lines for clarity. The horizontal dashed lines refer to equilibrium lattice constants under zero electric field.

## Sample deposition

The $La_{0.67}Sr_{0.33}MnO_3$ (LSMO) and $Hf_{0.5}Zr_{0.5}O_2$ (HZO) layers were deposited sequentially on (110)-oriented LAO, LSAT and STO single crystalline substrates by pulsed laser deposition (AdNano Corp.) both at 750 °C under 100 mTorr oxygen pressure. HZO and LSMO polycrystalline targets were all synthesized by conventional solid-state reaction. The ablation was performed with a pulsed 248 nm output from a PLD20 KrF excimer laser (Excimer, China) with an energy density of 1.5 J/cm² on the targets and a repetition rate of 2 Hz. In this work, the LSMO electrode is kept at 20 nm in thickness, and the HZO films are 10 nm in thickness.

## Sample characterization

The crystal structure of the HZO and LSMO layers was examined by XRD using the same Brucker D8 Discover diffractometer. Switching spectroscopy PFM measurements were performed on the bare HZO film surface at room temperature with a Pt/Ir coated cantilever probe (Nanoworld EFM) using an Asylum Research Cypher ES atomic force microscope, while the LSMO electrode was grounded.

## Reporting summary

Further information on research design is available in the Nature Portfolio Reporting Summary linked to this article.

## Data availability

The source data for Figs. 2–6 in this study are provided in the Source Data file. Source data are provided with this paper.

## Code availability

All DFT calculations were performed with VASP, which is proprietary software for which the Yang's lab owns a license.

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

## Acknowledgements

The authors thank the National Key R&D Program of China (Grant Nos. 2020YFA0711504 and 2022YFB3807601), and the National Science Foundation of China (Grants No. 12274201, No. 52232001, No. 51721001, No. 52003117, No. 92163210). They are grateful to the Program for Innovative Talents and Entrepreneurs in Jiangsu (JSSCTD202101) and the HPCC resources of Nanjing University for the calculations. L.B. thanks the Vannevar Bush Faculty Fellowship (VBFF) Grant no. N00014-20-1-2834 from the Department of Defense, Award No. DMR-1906383 from the National Science Foundation AMASE-i Program (MonArk NSF Quantum Foundry), Award No. FA9550-23-1-0500 from the U.S. Department of Defense under the DEPSCoR program and the Defense Advanced Research Projects Agency Defense Sciences Office (DSO) Program: Accelerating discovery of Tunable Optical Materials (ATOM) under Agreement no. HR00112390142.

## Author contributions

H.C. and P.J. contributed to this work equally. Y.Y. and D.W. conceived and designed the research. H.C. and J.W. performed the theoretical calculations supervised by Y.Y. and L.B. Samples were prepared by P.J. and M.Q., supervised by D.W. and Y.D. The experimental and theoretical results are analyzed by H.C., P.J., Y.D., J.L., L.B., D.W., and Y.Y. All authors followed the development of the research, discussed the results, and contributed to the preparation of the manuscript.

## Competing interests

The authors declare no competing interests.
