## [Peer Review File · Nature Communications]

Tunable and parabolic piezoelectricity in hafnia under epitaxial strainREVIEWER COMMENTS

Reviewer #1 (Remarks to the Author):

This manuscript reports on the tunable and parabolic piezoelectricity of HfO₂ under epitaxial strain. The findings are interesting, and the manuscript is well organized. There doesn't seem to be any major problems with the theoretical calculations. I just want to give my opinion that it is necessary to present more research results than experimental results. Overall, I would recommend that this manuscript be accepted after revision.

Reviewer #2 (Remarks to the Author):

This work investigates the piezoelectric coefficient of hafnia, combining theoretical calculations and experimental results. It claims the special tunable and parabolic piezoelectricity in hafnia under epitaxial strain. The paper is overall interesting, but there are still some issues to be resolved.

1. On Page 4, the paper claims that the monoclinic *P₂₁c* phase can be omitted in theoretical calculations because experimentally it is metastable for films below 30 nm thickness. This is obviously incorrect. Actually, the monoclinic phase is very robust and is still the thermodynamically most stable phase for thin films, except for some special circumstances. Without annealing, the ALD-grown hafnia film consists of extremely small fine grains, where the surface energy effect could render the tetragonal phase more stable (not the *Pca₂₁* phase). After annealing, the grains expand and monoclinic phase and *Pca₂₁* phase could be more stable, after cooling. There are a myriad of papers reporting the coexistence of monoclinic phase and *Pca₂₁* phase in ferroelectric hafnia for 10 nm thickness and below.

2. Since the m-phase is thermodynamically too stable, and the *Pca₂₁* phase is metastable, the calculations should reveal this fact because the m-phase should show much lower energy than the orthorhombic phase, especially under tensile strain. The key point of this paper is that the piezoelectric coefficient could change sign under tensile strain, but the m-phase is further stabilized under tensile strain, for it owns a larger cell volume. How could the effect be confirmed through calculations, provided that the most stable m-phase is excluded, especially under tensile strain?

3. On Page 9, the o-phase (111) peak is identified from the XRD patterns. Actually it greatly spreads and it is hard to identify its exact location. It turns out that the peak cannot simply be attributed to o-phase. Other phases also possess (111) peaks and probably also reside near that region.

4. The experimental part shows distinct piezoelectric effects on various substates. It is more desirable to envisage whether one can achieve dynamic transition between positive and negative piezoelectric coefficients in the same device. Therefore some theoretical design is needed.

5. In the Abstract, the authors emphasize that the unusual piezoelectric effect in hafnia is due to the chemical coordination of active oxygen atoms. However, I did not find more information or discussion in the article. Please provide more explanations or analysis.

6. Regarding the selection of research objectives, the author selected three phases of (111) oriented HfO_2 , namely $P4_2/nmc$ and $P4_2/mn$, in the article. The rhombohedral ferroelectric phase of hafnia, however, is also important. After all, the (111) oriented HfO_2 itself is close to the rhombohedral phase ($R3m$), and it has also been experimentally confirmed (Nature Materials 17, 1095 (2018)). More recently, the rhombohedral phase $\text{Hf}(\text{Zr})_{1+x}\text{O}_2$ has been confirmed to show low coercive field (Science 381, 558 (2023)). The rhombohedral phase is suggested to be considered in the calculations.

Reviewer #3 (Remarks to the Author):

The manuscript "Tunable parabolic piezoelectricity in hafnia under epitaxial strain" by Cheng et al., explores the positive and negative sign of the piezoelectric coefficient in ferroelectric HfO_2 thin films under epitaxial strain engineering. In the meantime, linear and non-linear behaviors of piezo responses are shown by theoretical calculations. The work helps the current understanding of piezoelectricity in HfO_2 -based ferroelectrics. The below questions should be addressed before the publication.

1. The manuscript references a prior work (Ref [21]) that also investigates the sign change in piezoelectric response under epitaxial strain, using first-principles calculations and experimental investigations on HfO_2 thin films. What's the main difference of this work from Ref[21]? Authors could highlight this in the manuscript.
2. All of calculations are based on (111)-oriented HfO_2 films. As shown in Fig. 4 (c), there is a notable difference in the intensity of (111) peaks at around $2\theta=30.2^\circ$ for HZO films of ~ 10 nm thickness on different substrates. Especially on LAO(110), the much lower intensity of HZO(111) peak indicate the other orientation or other phase existing. Could authors provide more specific evidence regarding the orientation of films on different substrates? It is worth considering whether the observed sign change and parabolic piezoelectric response remain consistent when films exhibit different orientations or combined orientations. The manuscript could benefit from discussing the potential impact of film orientation on these properties.
3. Authors employ HfO_2 in their DFT calculations, while the experimental results are presented for $\text{Hf}_{0.5}\text{Zr}_{0.5}\text{O}_2$ (HZO). It is known that different elements or doping can affect polarization a lot, thus the displacement of active oxygen atoms, which determine the piezoelectric response. The authors may include DFT calculations for HZO and provide its piezoelectric behavior.
4. The ferroelectricity is typically observed in HfO_2 films of around or below 10nm thickness, which exhibit the negligible piezoresponse. Could author provide some perspective about the potential applications of this materials with limited piezoelectricity in practical settings?
5. In Fig.5, authors employ MD simulations to illustrate the piezoelectric response of HfO_2 films under different epitaxial strains, which corresponds to the piezoelectricity observed in DFT calculations. I am curious whether the frequency of electric field will influence the piezoelectric behavior of films?

Reviewer #1 had the following comments in his/her report:

“This manuscript reports on the tunable and parabolic piezoelectricity of HfO₂ under epitaxial strain. The findings are interesting, and the manuscript is well organized. There doesn't seem to be any major problems with the theoretical calculations. I just want to give my opinion that it is necessary to present more research results than experimental results. Overall, I would recommend that this manuscript be accepted after revision.”

Our answer: We appreciate the positive statement of “The findings are interesting, and the manuscript is well organized. There doesn't seem to be any major problems with the theoretical calculations.” and “Overall, I would recommend that this manuscript be accepted after revision.”

Following Reviewer #1’s suggestion, we did more experiments and calculations.

We performed additional experiments for which the HZO thin films were deposited on LAO substrates buffered with LSMO in various thickness. The HZO film on top of the LSMO buffer layer can experience different in-plane strains, from compressive to tensile, as the strained LSMO buffer relaxes with increasing thickness. In this way, the piezoelectric characteristic of the HZO film can be studied as a function of in-plane strain with the other parameters being identical.

We added these results as Fig. 5 (Fig. R1 below), and a corresponding discussion in the second paragraph on page 10: “Consequently, the buffer LSMO layer with different thicknesses which is distinctly relaxed on the LAO substrate provides different epitaxial strains for HZO thin films. A series of HZO/LSMO heterostructures with identical thicknesses (10 nm) of HZO and different thicknesses of LSMO (20-66 nm) were synthesized on LAO substrates. XRD θ - 2θ scans of these samples are displayed in Fig. 5a. The peak shift of LSMO with increasing thickness from 20 to 66 nm indicates the gradual strain relaxation of the interface LSMO layer. Figures 5(c-e) show reciprocal space mappings around the (310) spot of LAO for HZO thin films deposited on 20-, 33-, and 66-nm-thick LSMO buffer layers, respectively. It is observed that the (310) spot of LSMO shifts toward the upper left with increasing thickness, indicating the relaxation with increasing in-plane (and concomitantly decreasing out-of-plane) lattice

constants. This allows us to continuously tune the epitaxial strain on HZO from a compressive to a tensile value, which arises from fully relaxed buffer layers. Figure 5b shows the PFM phase loops of the HZO films deposited on 20-, 33- and 66-nm-thick LSMO buffered LAO substrates. For the thinner buffer of 20 nm and 33 nm, the PFM shows a negative sign for the piezoelectric coefficient, while for the 66 nm buffer of LSMO (which is fully relaxed and provides a tensile strain for HZO), the piezoelectric coefficient adopts a positive sign.”

Fig. R1. [111]-oriented HZO films grown on (110) perovskite substrates with different thicknesses of LSMO buffer layers. **a**, XRD θ - 2θ patterns of HZO films deposited on (110)-oriented LAO with 20-, 33- and 66-nm-thick LSMO buffer layers. **b**, PFM phase loops of the HZO films deposited on 20-, 33- and 66-nm-thick LSMO buffered LAO substrates. **c**, **d**, **e**, Reciprocal space mappings around the (310) spot of HZO/LSMO/LAO with (c) 20-, (d) 33- and (e) 66-nm-thick LSMO buffer layers.

We conducted additional calculations for which the monoclinic phase HfO_2 under different epitaxial strain is considered (as shown in Fig. S1), and the piezoelectric coefficients of $\text{Hf}_{0.5}\text{Zr}_{0.5}\text{O}_2$ (not only HfO_2) are calculated (as shown in Fig. S6). The corresponding discussion of these additional calculations of Fig. S1 and S6 are added on pages 1 and 8, respectively, in supplemental materials.

Reviewer #2 had the following comments in his/her report:

“This work investigates the piezoelectric coefficient of hafnia, combining theoretical calculations and experimental results. It claims the special tunable and parabolic piezoelectricity in hafnia under epitaxial strain. The paper is overall interesting, but there are still some issues to be resolved.”

Our answer: We appreciate the positive statement “The paper is overall interesting”.

(1) *“On Page 4, the paper claims that the monoclinic $P2_1/c$ phase can be omitted in theoretical calculations because experimentally it is metastable for films below 30 nm thickness. This is obviously incorrect. Actually, the monoclinic phase is very robust and is still the thermodynamically most stable phase for thin films, except for some special circumstances. Without annealing, the ALD-grown hafnia film consists of extremely small fine grains, where the surface energy effect could render the tetragonal phase more stable (not the $Pca2_1$ phase). After annealing, the grains expand and monoclinic phase and $Pca2_1$ phase could be more stable, after cooling. There are a myriad of papers reporting the coexistence of monoclinic phase and $Pca2_1$ phase in ferroelectric hafnia for 10 nm thickness and below.”*

Our answer: We thank Referee #2 for pointing this out. We rewrote this sentence Referee # 2 mentioned in the first paragraph on page 4 as: “We consider the (111)- HfO_2 films in monoclinic $P2_1/c$ phase, orthorhombic $Pca2_1$ phase, tetragonal $P4_2/nmc$ phase, and another orthorhombic $Pmn2_1$ phase, and rhombohedral phases, as these phases were experimentally synthesized in thin films³²⁻⁴⁸.”

New references [47, 48] are also added:

- 47 Lyu, J., Fina, I., Solanas, R., Fontcuberta, J. & Sánchez, F. Growth window of ferroelectric epitaxial $Hf_{0.5}Zr_{0.5}O_2$ thin films. *ACS Appl. Electron. Mater.* **1**, 220-228 (2019).
- 48 Jiao, P. *et al.* Flexoelectricity-stabilized ferroelectric phase with enhanced reliability in ultrathin La: HfO_2 films. *Appl. Phys. Rev.* **10**, 031417 (2023).

(2) *“Since the m -phase is thermodynamically too stable, and the $Pca2_1$ phase is metastable, the calculations should reveal this fact because the m -phase should show much lower energy than the orthorhombic phase, especially under tensile strain. The key point of this paper is that the piezoelectric coefficient could change sign under tensile strain, but the m -phase is further stabilized under tensile strain, for it owns a larger cell volume. How could the effect be confirmed through calculations, provided that the most stable m -phase is excluded, especially under tensile strain?”*

Our answer: This is a valid point. We performed additional calculations of the energy

of the m -phase under different strains, as shown in the phase diagram of Fig. R2 below. As reviewer #2 mentioned, the m -phase shows lower energy than the orthorhombic phase under tensile strain, as similar to the recent work [Phys. Rev. B 108, L060102 (2023)]. Note that our experiments show that HZO under LSAT (with strain 2.4%) and STO (with strain 3.4%) substrates are ferroelectric phases. We conclude that the interface effects (between HZO and substrate) and surface effects make the ferroelectric phase stable, though the ferroelectric phase has higher energy than m -phase. Since surface and interface effects are difficult to include in our calculations, we therefore focus on the properties of the o -phase and do not consider the m -phase in the manuscript.

Fig. R2. Energies of different phases (including monoclinic $P2_1/c$ phase) as a function of the epitaxial strain.

We added sentences in the second paragraph on page 4: “On the other hand, for tensile strains larger than 1.5%, the energy of the $P2_1/c$ -like phase has lower energy than the $Pca2_1$ -like phase and becomes the ground state (see Fig. S1). The energy difference between the $P2_1/c$ -like phase and the $Pca2_1$ -like phase under tensile strain is comparable to that of these phases in bulk⁵⁰. In our experiments (see Figs. 4 and 5), $Pca2_1$ -like rather than $P2_1/c$ -like phase is grown successfully, though the $Pca2_1$ -like phase is higher energy than $P2_1/c$ -like phase. This may come from surface and/or

interface effects which make the $Pca2_1$ -like phase more stable (see Figs. 4 and 5). We therefore focus on the piezoelectric response of $Pca2_1$ -like phase from -3% to 4.5%.”

We also added Fig. S1 and the corresponding discussion in the Supplemental Material.

New reference [50] is also added:

50 Zhu, T., Deng, S. & Liu, S. Epitaxial ferroelectric hafnia stabilized by symmetry constraints. *Phys. Rev. B* **108**, L060102 (2023).

(3) “On Page 9, the o -phase (111) peak is identified from the XRD patterns. Actually it greatly spreads and it is hard to identify its exact location. It turns out that the peak cannot simply be attributed to o -phase. Other phases also possess (111) peaks and probably also reside near that region.”

Our answer: We thank the reviewer's suggestions. Complex phase competition indeed occurs in HfO_2 -based thin films. However, in epitaxial films, it is mainly the competition between two phases, these are the orthorhombic phase (sometimes distorted) and the monoclinic phase. Although the peak is broad, the peak position can be clearly identified at 2θ around 30.2° , coincident with the Bragg angle of the o -HZO (111) reflection. Liu *et al.* considered epitaxial films of different orientations and strains and evaluated by DFT calculations the energy of orthorhombic phases in comparison with the energy of the monoclinic phase. They concluded that the orthorhombic phases were more stable than the monoclinic phase in (111) films in a wide range of strain [Phys. Rev. Mater. 3, 054404 (2019)]. Furthermore, reflections of the monoclinic phase, ($\bar{1}11$) at a 2θ of around 28.5° and (002) at a 2θ of around 35° , commonly observed in polycrystalline films, are not detected in our samples. [Appl. Phys. Lett. 104, 072901 (2014); Nat. Mater. 17, 1095 (2018)]. Therefore, the peak at 2θ around 30.2° can be safely identified as an o -HZO (111) diffraction, as commonly reported in the literature on epitaxial HZO films. [Adv. Funct. Mater. 33, 2209925 (2022); Nat. Commun. 14, 1780 (2023)].

(4) “The experimental part shows distinct piezoelectric effects on various substates. It is more desirable to envisage whether one can achieve dynamic transition between positive and negative piezoelectric coefficients in the same device. Therefore some theoretical design is needed.”

Our answer: This is a good point. In addition to using various substrates, changing the thickness of buffer layers is another way to distinguish piezoelectric effects. We performed additional experiments to achieve this transition between positive and negative piezoelectric coefficients by tuning the thickness of buffer layers. Please see Figure 5 and the corresponding discussion on Page 10.

In order to achieve a dynamic transition between positive and negative piezoelectric coefficients, one way is to transfer the HfO₂-base thin film to the organic substrate polyethylenenaphthalate (PEN) and use epoxy as the glue to transfer strain to the HfO₂-base thin film [Ding, et al., *Adv. Funct. Mater.* **33**, 2213725 (2023)]. Without strain, the piezoelectric coefficient of the HfO₂ thin film on PEN is negative (see Fig. 2d and Fig. S5), while it is positive when the HfO₂ thin film is under tensile strain by stretching epoxy.

We added the statement “In order to achieve a dynamic transition between positive and negative piezoelectric coefficients, one way is to transfer the HfO₂-base thin film to the organic substrate polyethylenenaphthalate (PEN) and use epoxy as the glue to transfer strain to the HfO₂-base thin film. Without strain, the piezoelectric coefficient of the HfO₂ thin film on PEN is negative (see Fig. 2d and Fig. S5), while it is positive when the HfO₂ thin film is under tensile strain by stretching epoxy.” in the second paragraph on page 11, to provide a design about achieving dynamic transition between positive and negative piezoelectric coefficients in the same device.

New reference [52] is also added:

52 Ding, Z. *et al.* Observation of uniaxial strain tuned spin cycloid in a freestanding BiFeO₃ film. *Adv. Funct. Mater.* **33**, 2213725 (2023).

(5) “*In the Abstract, the authors emphasize that the unusual piezoelectric effect in hafnia is due to the chemical coordination of active oxygen atoms. However, I did not find more information or discussion in the article. Please provide more explanations or analysis.*”

Our answer: This is another good point. The chemical coordination of the oxygen atoms in HfO₂ is different from that of prototypical ferroelectrics such as PbTiO₃. In HfO₂, the oxygen atoms can be divided into two categories, one kind of oxygen atom

induces polarization and the other does not contribute to polarization. Furthermore, the oxygen atoms induced polarization can be classified into four types which have different contributions to piezoelectric effects (see Fig. 2d and Fig. 3 with the discussion on Pages 6-8).

To make the role of chemical coordination of active oxygen atoms on the piezoelectric effect more clear, we added the sentences “The displacement of O_1 under large compressive or tensile epitaxial strains dominates the total displacement.” and “The displacement of the four types of O atoms is very small around the epitaxial strain of 1.5%, which means that their contribution to polarization change is very small, resulting in minimal changes in overall polarization.” in the first paragraph on page 7, and also added sentences of “Overall, the unusual piezoelectric effect in hafnia is due to the competitive relationship of the four kinds of oxygen atoms on the displacement under epitaxial strain.” in the first paragraph on page 8.

(6) “Regarding the selection of research objectives, the author selected three phases of (111) oriented HfO_2 , namely $Pca2_1$, $P4_2/nmc$ and $Pmn2_1$, in the article. The rhombohedral ferroelectric phase of hafnia, however, is also important. After all, the (111) oriented HfO_2 itself is close to the rhombohedral phase ($R3m$), and it has also been experimentally confirmed (*Nature Materials* 17, 1095 (2018)). More recently, the rhombohedral phase $Hf(Zr)_{1+x}O_2$ has been confirmed to show low coercive field (*Science* 381, 558 (2023)). The rhombohedral phase is suggested to be considered in the calculations.”

Our answer: We thank Reviewer #2 for this suggestion. We performed additional calculations on rhombohedral phases such as the $R3m$ and $R3$. In the calculations of [111]-oriented films, the $R3m$ phase relaxes to a $P4_2/nmc$ -like phase, and the $R3$ phase relaxes to a $Pca2_1$ -like phase. These results are consistent with previous first-principles calculations where the rhombohedral phases have much higher energy than the orthorhombic phase [*Nat. Mater.* 17, 1095-1100 (2018); *Phys. Rev. Lett.* 125, 257603 (2020)].

Therefore, we added the sentences “The $R3m$ and $R3$ phases^{38,49} are also considered in our calculations. However, the $R3m$ phase relaxes to a $P4_2/nmc$ -like phase, and the $R3$ phase turns to a $Pca2_1$ -like phase under the strain considered here.” in the first

paragraph on page 4.

New reference [49] is added:

- 49 Wei, Y. *et al.* A rhombohedral ferroelectric phase in epitaxially strained $\text{Hf}_{0.5}\text{Zr}_{0.5}\text{O}_2$ thin films. *Nat. Mater.* **17**, 1095-1100 (2018).

Reviewer #3 had the following comments in his/her report:

“The manuscript ‘Tunable and parabolic piezoelectricity in hafnia under epitaxial strain’ by Cheng et al., explores the positive and negative sign of the piezoelectric coefficient in ferroelectric HfO_2 thin films under epitaxial strain engineering. In the meantime, linear and non-linear behaviors of piezo responses are shown by theoretical calculations. The work helps the current understanding of piezoelectricity in HfO_2 -based ferroelectrics. The below questions should be addressed before the publication.”

Our answer: We thank Reviewer #3 for his/her positive statement that *“The work helps the current understanding of piezoelectricity in HfO_2 -based ferroelectrics”*.

(1) *“The manuscript references a prior work (Ref [21]) that also investigates the sign change in piezoelectric response under epitaxial strain, using first-principles calculations and experimental investigations on HfO_2 thin films. What’s the main difference of this work from Ref[21]? Authors could highlight this in the manuscript.”*

Our answer: We thank the reviewer's suggestions. Reference [21] indeed investigated the sign change in the piezoelectric response under epitaxial strain. This paper first predicted that the piezoelectric effect is negative at tensile and small compressive [001] epitaxial strains while the piezoelectric coefficient is positive at large compressive [001] epitaxial strains. Reference [21] also provided the negative piezoelectric PFM phase loop of La: HfO_2 thin film. In our manuscript, we considered [111] epitaxial strain which usually occurs in thin films grown by PLD methods. The transition of sign of piezoelectric effect was observed by calculations and experiments at different [111] epitaxial strained thin films. Furthermore, we found nonlinear and even parabolic piezoelectric behavior for the first time.

To highlight these differences, we added the sentences “Different from the previous work²¹, our calculations are conducted on (111)-oriented thin films which are favorable in experiments with compatibility existing between the substrates and the thin films.”,

“Significantly, we elucidate the piezoelectric phenomenon by examining atomic displacements at the microscopic level. Our calculations agree with the experimental results.” in the second paragraph on page 3, and also added “Notably, the parabolic piezoelectricity is unveiled in hafnia at small tensile epitaxial strain, thereby advancing the current understanding of the field of piezoelectricity.” in the first paragraph on page 6.

(2) *“All of calculations are based on (111)-oriented HfO₂ films. As shown in Fig. 4 (c), there is a notable difference in the intensity of (111) peaks at around $2\theta=30.2^\circ$ for HZO films of ~10 nm thickness on different substrates. Especially on LAO(110), the much lower intensity of HZO(111) peak indicate the other orientation or other phase existing. Could authors provide more specific evidence regarding the orientation of films on different substrates? It is worth considering whether the observed sign change and parabolic piezoelectric response remain consistent when films exhibit different orientations or combined orientations. The manuscript could benefit from discussing the potential impact of film orientation on these properties.”*

Our answer: This is another good point. First of all, the monoclinic phase can be ruled out because the $(\bar{1}11)$ diffraction at 2θ around 28.5° and the (002) diffraction at 2θ around 35° , commonly observed in polycrystalline films, are not detected in our samples [Appl. Phys. Lett. 104, 072901 (2014)]. The different diffraction intensity of the HZO (111) peak on (110)-oriented LAO, LSAT, and STO substrates may be ascribed to different microstructures in the HZO films. To rule out the influence of the microstructure, we investigated the piezoelectric characteristics of a series of HZO/LSMO/LAO samples, with the HZO film kept at the same thickness (10 nm) but with the LSMO buffer layer adopting various thicknesses (20-66 nm). The HZO film on top of the LSMO buffer layer can thus experience different in-plane strains, from compressive to tensile, as the strained LSMO buffer relaxes with increasing thickness. In this way, the piezoelectric characteristic of the HZO film can be studied as a function of in-plane strain with the HZO films in the same crystalline quality. The results are in excellent agreement with those on different substrates. Brief discussions have been included in the revised manuscript (second paragraph on page 10).

(3) “Authors employ HfO_2 in their DFT calculations, while the experimental results are presented for $\text{Hf}_{0.5}\text{Zr}_{0.5}\text{O}_2$ (HZO). It is known that different elements or doping can affect polarization a lot, thus the displacement of active oxygen atoms, which determine the piezoelectric response. The authors may include DFT calculations for HZO and provide its piezoelectric behavior.”

Our answer: We thank Reviewer #3 for this suggestion. We performed additional calculations for $\text{Hf}_{0.5}\text{Zr}_{0.5}\text{O}_2$ and found very similar piezoelectric behavior to that for HfO_2 . As shown in Fig. R3 below, the negative first-order piezoelectric coefficient at compressive strains and small tensile strains transforms into a positive coefficient at tensile strains larger than 1%. The second-order coefficient is always negative and becomes very large at tensile strains larger than 3%.

We added the results for $\text{Hf}_{0.5}\text{Zr}_{0.5}\text{O}_2$ as Fig. S6 in the Supplemental Material, and the corresponding discussion “Hf and Zr belong to the same column of the Periodic Table and thus have very similar chemical properties. In addition, due to the lanthanide contraction, the ion radii of Hf and Zr are very close to each other. Fig. S6 shows the piezoelectric behavior of $\text{Hf}_{0.5}\text{Zr}_{0.5}\text{O}_2$ which is similar to that of HfO_2 (see Fig. 2c in main text). The negative first-order piezoelectric coefficient at compressive strains and small tensile strains transforms into a positive coefficient at tensile strains larger than 1%. The second-order coefficient is always negative and becomes very large at tensile strains larger than 3%.” in Section IV of the Supplemental Material.

Fig. R3. The calculated linear piezoelectric coefficients e_{33} and quadratic coefficient B_{333} of $\text{Hf}_{0.5}\text{Zr}_{0.5}\text{O}_2$ as functions of epitaxial strain.

(4) “The ferroelectricity is typically observed in HfO_2 films of around or below 10 nm

thickness, which exhibit the negligible piezoresponse. Could author provide some perspective about the potential applications of this materials with limited piezoelectricity in practical settings?”

Our answer: We thank the Referee for the valuable suggestion. Piezoelectric thin films are very promising for their application in sensory, actuation systems, energy harvesting, as well as medical and acoustic transducers. For thicknesses below 10 nm and due to its nontoxic nature, HfO₂-based piezoelectric films can also be used in medical devices for diagnoses inside the human body and are good candidates for implantable cardio-mechanical electric sensor applications with limited piezoelectricity. Also, as HfO₂ can be grown on silicon and is compatible with the electronics industry, HfO₂ is a good candidate for piezoelectric microelectromechanical systems (MEMS) on the nanoscale.

We added the statement “For the nanometric thicknesses and nontoxic nature, HfO₂-based piezoelectric film can be used in medical devices for diagnoses inside the human body, and are good candidates for implantable cardio-mechanical electric sensor applications with limited piezoelectricity. Also, as HfO₂ thin films can be grown on silicon and are compatible with the electronics industry, they are also good candidates for piezoelectric microelectromechanical systems (MEMS) in nanoscale.” in the third paragraph on page 12, to discuss the possible application for HfO₂-base piezoelectric thin films.

(5) “In Fig. 5, authors employ MD simulations to illustrate the piezoelectric response of HfO₂ films under different epitaxial strains, which corresponds to the piezoelectricity observed in DFT calculations. I am curious whether the frequency of electric field will influence the piezoelectric behavior of films?”

Our answer: This is a valid point. The piezoelectric response comes from atomistic displacements (lattice change) under electric field. The time scale of atomic (molecular) dynamics is from femtosecond (fs, 10⁻¹⁵ second) to picosecond (ps, 10⁻¹² second), which corresponds to a terahertz (THz) frequency. Usually, the electric field has much lower frequency than THz, and the piezoelectric behavior is independent of the electric field frequency. On the other hand, if the piezoelectric material is a ferroelectric relaxor, the piezoelectric behavior would be dependent on the frequency of electric field as the complex microscopic structures of the relaxor.

Summary of changes:

The revisions (sentences added or revised) are marked in red on the main text and the Supplementary Material.

- We added Fig. 5, and the corresponding discussion in the second paragraph on page 10: “Consequently, the buffer LSMO layer with different thicknesses which is distinctly relaxed on the LAO substrate provides different epitaxial strains for HZO thin films. A series of HZO/LSMO heterostructures with identical thicknesses (10 nm) of HZO and different thicknesses of LSMO (20-66 nm) were synthesized on LAO substrates. XRD θ -2 θ scans of these samples are displayed in Fig. 5a. The peak shift of LSMO with increasing thickness from 20 to 66 nm indicates the gradual strain relaxation of the interface LSMO layer. Figures 5(c-e) show reciprocal space mappings around the (310) spot of LAO for HZO thin films deposited on 20-, 33-, and 66-nm-thick LSMO buffer layers, respectively. It is observed that the (310) spot of LSMO shifts toward the upper left with increasing thickness, indicating the relaxation with increasing in-plane (and concomitantly decreasing out-of-plane) lattice constants. This allows us to continuously tune the epitaxial strain on HZO from a compressive to a tensile value, which arises from fully relaxed buffer layers. Figure 5b shows the PFM phase loops of the HZO films deposited on 20-, 33- and 66-nm-thick LSMO buffered LAO substrates. For the thinner buffer of 20 nm and 33 nm, the PFM shows a negative sign for the piezoelectric coefficient, while for the 66 nm buffer of LSMO (which is fully relaxed and provides a tensile strain for HZO), the piezoelectric coefficient adopts a positive sign.”, in response to comment from Referee #1.
- We rewrote the sentence in the first paragraph on page 4: “We consider the (111)-HfO₂ films in monoclinic $P2_1/c$ phase, orthorhombic $Pca2_1$ phase, tetragonal $P4_2/nmc$ phase, and another orthorhombic $Pmn2_1$ phase, and rhombohedral phases, as these phases were experimentally synthesized in thin films³²⁻⁴⁸.”, in response to comment 1 from Referee #2.
- References [47,48] are added to the main text, in response to comment 1 from Referee #2.
- We added the sentences in the second paragraph on page 4: “On the other hand, for tensile strains larger than 1.5%, the energy of the $P2_1/c$ -like phase has lower energy than the $Pca2_1$ -like phase and becomes the ground state (see Fig. S1). The energy difference between the $P2_1/c$ -like phase and the $Pca2_1$ -like phase under tensile strain is comparable to that of these phases in bulk⁵⁰. In our experiments (see Figs. 4 and 5), $Pca2_1$ -like rather than $P2_1/c$ -like phase is grown successfully, though the $Pca2_1$ -like phase is higher energy than $P2_1/c$ -like phase. This may come from

surface and/or interface effects which make the $Pca2_1$ -like phase more stable (see Figs. 4 and 5). We therefore focus on the piezoelectric response of $Pca2_1$ -like phase from -3% to 4.5%.”, in response to comment 2 from Referee #2.

- We added Fig. S1 and the sentences in the first paragraph on Page 3 in the Supplemental Material: “Figure S1 shows energies of different phases with epitaxial strain of (111)-oriented hafnia. For tensile strains larger than 1.5%, the energy of the $P2_1/c$ -like phase is the lowest, meanwhile, the energy difference between the $P2_1/c$ -like phase and the $Pca2_1$ -like phase is comparable to that of the bulk, which is similar to the recent calculations of Ref. [6].”, in response to comment 2 from Referee #2.
- Reference [50] is added to the main text, which is the Reference [6] in the Supplemental Material, in response to comment 2 from Referee #2.
- We added the statement “In order to achieve a dynamic transition between positive and negative piezoelectric coefficients, one way is to transfer the HfO_2 -base thin film to the organic substrate polyethylenephthalate (PEN) and use epoxy as the glue to transfer strain to the HfO_2 -base thin film. Without strain, the piezoelectric coefficient of the HfO_2 thin film on PEN is negative (see Fig. 2d and Fig. S5), while it is positive when the HfO_2 thin film is under tensile strain by stretching epoxy.” in the second paragraph on page 11, in response to comment 4 from Referee #2.
- Reference [52] is added to the main text, in response to comment 4 from Referee #2.
- We added the sentences “The displacement of O_1 under large compressive or tensile epitaxial strains dominates the total displacement.” and “The displacement of the four types of O atoms is very small around the epitaxial strain of 1.5%, which means that their contribution to polarization change is very small, resulting in minimal changes in overall polarization.” in the first paragraph on page 7, and also added sentences of “Overall, the unusual piezoelectric effect in hafnia is due to the competitive relationship of the four kinds of oxygen atoms on the displacement under epitaxial strain.” in the first paragraph on page 8, in response to comment 5 from Referee #2.
- We added the sentences “The $R3m$ and $R3$ phases^{38,49} are also considered in our calculations. However, the $R3m$ phase relaxes to a $P42/nmc$ -like phase, and the $R3$ phase turns to a $Pca2_1$ -like phase under the strain considered here.” in the first paragraph on page 4, in response to comment 6 from Referee #2.
- Reference [49] is added to the main text, in response to comment 6 from Referee #2.

- We added the sentences “Different from the previous work²¹, our calculations are conducted on (111)-oriented thin films which are favorable in experiments with compatibility existing between the substrates and the thin films.”, “Significantly, we elucidate the piezoelectric phenomenon by examining atomic displacements at the microscopic level. Our calculations agree with the experimental results.” in the second paragraph on page 3, and also added “Notably, the parabolic piezoelectricity is unveiled in hafnia at small tensile epitaxial strain, thereby advancing the current understanding of the field of piezoelectricity.” in the first paragraph on page 6, in response to comment 1 from Referee #3.
- We added Fig. S6 and the corresponding discussion “Hf and Zr belong to the same column of the Periodic Table and thus have very similar chemical properties. In addition, due to the lanthanide contraction, the ion radii of Hf and Zr are very close to each other. Fig. S6 shows the piezoelectric behavior of $\text{Hf}_{0.5}\text{Zr}_{0.5}\text{O}_2$ which is similar to that of HfO_2 (see Fig. 2c in main text). The negative first-order piezoelectric coefficient at compressive strains and small tensile strains transforms into a positive coefficient at tensile strains larger than 1%. The second-order coefficient is always negative and becomes very large at tensile strains larger than 3%.” in Section IV of the Supplemental Material, in response to comment 3 from Referee #3.
- We added the statement “For thicknesses below 10 nm and due to its nontoxic nature, HfO_2 -based piezoelectric film can be used in medical devices for diagnoses inside the human body, and are good candidates for implantable cardio-mechanical electric sensor applications with limited piezoelectricity. Also, as HfO_2 thin films can be grown on silicon and are compatible with the electronics industry, they are also good candidates for piezoelectric microelectromechanical systems (MEMS) in nanoscale.” in the third paragraph on page 12, in response to comment 4 from Referee #3.

REVIEWERS' COMMENTS

Reviewer #2 (Remarks to the Author):

The authors have resolved the major issue, which is regarding the estimation of the energy in the monoclinic phase. The other few points are basically resolved.

Reviewer #3 (Remarks to the Author):

Authors answered my questions, the revised manuscript can be accepted.

Reviewer #2 had the following comments in his/her report:

“The authors have resolved the major issue, which is regarding the estimation of the energy in the monoclinic phase. The other few points are basically resolved.”

Our answer: We are pleased to have dealt with all the issues raised by Reviewer #2 with satisfaction and appreciate the referee recommending our work for publication in Nature Communications.

Reviewer #3 had the following comments in his/her report:

“Authors answered my questions, the revised manuscript can be accepted.”

Our answer: We sincerely thank the referee for recommending our work to be published in Nature Communications.

Summary of changes:

There are no changes in the revisions.